

# Multi-class waste segregation using computer vision and robotic arm

Jayanti Lahoti, Jathin Sn, M. Vamshi Krishna, Mallika Prasad, Rajeshwari BS, Namratha Mysore and Jyothi S. Nayak

B.M.S. College of Engineering, Bangalore, India

## ABSTRACT

Waste segregation is an essential aspect of a smoothly functioning waste management system. Usually, various recyclable waste types are disposed of together at the source, and this brings in the necessity to segregate them into their categories. Dry waste needs to be separated into its own categories to ensure that the proper procedures are implemented to treat and process it, which leads to an overall increased recycling rate and reduced landfill impact. Paper, plastics, metals, and glass are just a few examples of the many dry waste materials that can be recycled or recovered to create new goods or energy. Over the past years, much research has been conducted to devise effective and productive ways to achieve proper segregation for the waste that is being produced at an ever-increasing rate. This article introduces a multi-class garbage segregation system employing the YOLOv5 object detection model. Our final prototype demonstrates the capability of classifying dry waste categories and segregating them into their respective bins using a 3D-printed robotic arm. Within our controlled test environment, the system correctly segregated waste classes, mainly paper, plastic, metal, and glass, eight out of 10 times successfully. By integrating the principles of artificial intelligence and robotics, our approach simplifies and optimizes the traditional waste segregation process.

## INTRODUCTION

Waste management is one of the most significant and crucial challenges that our nation is currently facing in the modern world. This problem is rapidly escalating out of control and posing a substantial risk to the environment as a result of the nonstop production of waste that occurs around the clock every single day. However, it is essential to debunk the widespread belief that recycling is a fruitless endeavor to make progress in this area. The majority of waste types have the potential to entirely transform or break down into harmless composites that can serve alternative purposes, and an astounding 85% of the waste that households produce is recyclable effectively. Household waste needs to be separated appropriately (*Sruthy et al., 2021*) into various categories to facilitate the streamlining of recycling procedures, which is the key to unlocking the opportunities presented by recycling. An intelligent waste management system, one that takes advantage of recent developments in technology to automatically verify and categorize waste with a level of accuracy that is

Corresponding author
Namratha Mysore,
namratham.cse@bmsce.ac.in

unmatched (*Dong, 2021*; *Adedeji & Wang, 2019*), has become an absolute necessity as the volume of waste produced continues to balloon to ever-greater proportions. By adopting the strategy of automation in the waste classification and segregation system (*Nandhini et al., 2019*), it is possible to reap a variety of benefits, such as increased productivity, enhanced accuracy, enhanced sustainability, significant cost savings, and improved health and safety conditions. The integration of artificial intelligence (AI) and robotics has emerged as a pioneering solution in this context, revolutionizing the field of waste management and providing a pathway to tackle the current waste crisis effectively (*Tasnim et al., 2022*).

Despite the undeniable potential for waste recycling, achieving widespread waste segregation continues to be a challenge. This is especially true in nations like India (*Sudha et al., 2016*), which have a large population and are experiencing rapid economic growth. The majority of the population is hesitant to adopt practices for waste segregation due to factors including a lack of adequate education, a scarcity of resources, and an inherent unwillingness on their part. As a result, creating an intelligent waste management system that makes use of the power of cutting-edge technology has become essential to overcoming these obstacles (*Bircanoğlu et al., 2018*). The separation of waste according to its inherent properties, such as dryness, wetness, and biodegradability, is becoming increasingly important in the effort to lessen the negative impact that is currently having on the environment.

In spite of the fact that the separation of wet and dry waste is already an accepted and common practice, further separating dry waste is of the utmost importance because it enables the waste to be optimally prepared for reuse, recycling, or any other form of recovery. Not only does an effective waste segregation process cut down on the amount of waste that is sent to landfills, but it also minimizes the amount of waste that is exposed to the air and water, which helps to protect the health of the environment as a whole. The manual sorting of waste on a massive scale has been the primary method of waste segregation for the majority of history. However, the volume of waste that is being produced currently exceeds the capacity of manual processes, so novel approaches are required to deal with the surplus of waste. As a result, researchers have put a lot of time and effort into automating waste segregation (*Ahmad, Khan & Al-Fuqaha, 2020*), and they have put forth a variety of strategies to increase the effectiveness of this process. In this effort to find better solutions for waste management, artificial intelligence, and robotics have emerged as two of the most promising areas of research. This combination of artificial intelligence and robotics presents a significant opportunity to bring about a revolution in the existing system of waste management by improving the precision and effectiveness of waste sorting procedures (*Usha & Mahesh, 2022*).

In light of these recent developments, this article presents a thorough analysis of the current state of the field by investigating a diverse range of research and initiatives that have been carried out in this area and proposes a multi-class garbage segregation system using computer vision and a robotic arm. The concepts of artificial intelligence and robotics have been utilized to make this process easier and more efficient. Further, the article highlights potential avenues for future enhancements while presenting the results that were achieved through the use of these innovative approaches.

The article is arranged as follows in the mentioned order: Literature Review, Proposed Methodology, System Implementation, Experimental Setup, Results and lastly followed by a Conclusion.

## RELATED WORK

*Narayanswamy, Rajak & Hasan (2022)* compared YOLO, CNN, and RCNN and concluded that YOLO is better at classifying multiple wastes in the same frame, which can be used when the budget is low and high accuracy is needed. The faster RCNN is the best classifier, which has the highest accuracy of 91% and a loss of 16% so far, and the algorithm requires high power and a lot of data.

*Zhou et al. (2021)* designed a self-made dataset of 4 categories, namely plastic bottles, glass bottles, metal cans, and cartoons. The trash detection network is built upon YOLOV4 with GhostNet as the backbone in an attempt to construct an improved YOLOV4. Improved YOLOV4 implied that training and inference times were faster while accuracy also increased. A dual-arm robot with mobility is integrated with the trained detection network for the convenience of trash identification and pickup.

*Padalkar, Pathak & Stynes (2021)* presented an object detection and scaling model for plastic waste sorting and detection of four types of plastic using the WaDaBa database. In data modeling and training, the Scaled-Yolov and EfficientDet were pre-trained with the COCO dataset, and the weights were fine-tuned on a plastic dataset. The final bounding box is exported as a JSON file and uploaded to Roboflow. Scaled-Yolov4-CSP has the highest accuracy of 97% compared to the other models.

*Sai Sushanth, Jenila Livingston & Agnel Livingston (2021)* presented manual segregation of waste by automating it using CNN. Image scraping is used in the project, which involves collecting images from the web and making a dataset out of them. The misclassified images are removed, and new images are added. Finally, accuracy, precision, and recall are calculated using true positives, true negatives, false positives, and false negatives. The most misclassified among the classes is glass, so more glass images can be added to enhance the model.

*Sheth et al. (2010)* aim to automate tasks that previously would have required manual intervention, such as sorting items using machine vision. A proximity sensor, a microcontroller, and a USB webcam are the electronic parts, with a robot arm and a conveyor belt being the mechanical parts. The robotic arm will pick and place the component according to color if it fits the requirement and this process will then be repeated as many times as required. The advantages of this system are its fast speed, reduced labor cost, and good repeatability.

*Thanawala, Sarin & Verma (2020)* proposed a voice-controlled robotic arm with 5 degrees of freedom for the automatic segregation of medical waste. It involves three modules speech-to-text module that uses Google Cloud's speech API for conversion of speech to text, A waste detection and classification module in which camera images are fed into the Yolov3 algorithm, and a pickup and place module that mainly uses two ROS packages, ROS TF, and ROS MoveIt.

*Chinnathurai et al. (2016)* aimed to build a robot (Recylebot) that segregates recyclable and non-recyclable wastes automatically. The system has four modules (*i.e.,* drivetrain, image acquisition system, image processing server, and human-machine interface). The drivetrain module is a motor control that uses UART from Raspberry Pi, and it also uses ultrasonic sensors to detect nearby objects. The image acquisition system is at the top of the robot, near the camera. The Image processing server is a remote module that uses MATLAB for image processing. The human-machine interface is a GUI for the human user.

## Proposed methodology for YOLO single shot detector interfaced with 5DOF robotic arm system

By fusing software and hardware elements, the methodology suggested in this article introduces a novel approach to waste segregation, leveraging the capabilities of the YOLOv5 classifier in conjunction with a 5DOF robotic arm.

A webcam serves as the primary sensor, capturing live video feeds of the waste materials presented to the system. The video stream is processed in real time, where it is split into individual frames. These frames provide the raw data necessary for object detection and classification. To train the object classification model, a robust dataset is essential. This dataset consists of numerous images of various annotated waste items. The quality and diversity of the dataset directly correlate with the subsequent accuracy of the object classification model. The model learns to identify and categorize waste items by recognizing patterns and features within the annotated images. It undergoes a rigorous training regime where its parameters are iteratively adjusted to minimize errors in object localization and classification. Upon completion of the training phase, the YOLOv5 model is deployed on a system-on-chip (SoC) platform, enabling the real-time processing of video frames. As each frame is analyzed, the model localizes and classifies objects within the frame, tagging them with the appropriate waste category. The classification result generated by the SoC is transmitted *via* serial communication to an Arduino microcontroller. This microcontroller interprets the data and converts it into actionable commands for the robotic arm's DC motors. The robotic arm, equipped with a five-degree-of-freedom manipulator and a pick-and-place module, executes these commands. The robotic arm, guided by the classification input, physically segregates the waste by picking up items and placing them into designated bins corresponding to their categories. This segregation process is targeted and precise, ensuring that each item is deposited in the correct bin.

## You Only Look Once

The You Only Look Once (YOLO) family is renowned for its single-shot detection capabilities. It is called a single-shot detector because it processes an entire image in one forward pass, unifying object localization and classification. This approach makes it significantly faster than multi-step methods, allowing for real-time object detection with decent accuracy. The underlying architecture in YOLO divides the image into a grid, and each grid cell is in charge of identifying objects in its assigned region. This significantly improves computational efficiency by processing the entire image in a single pass.

Our methodology utilizes YOLOv5 because it can achieve greater accuracy and shows improved generalization across a wider range of waste categories due to its efficient architecture. YOLOv5 was chosen over its successors (like YOLOv8) because of its lesser memory utilization and its compatibility with devices like the Pi3B+ (*Kumar et al., 2021*). This framework enables the identification of different types of waste and their locations, along with the detection of their boundaries. YOLOv5's architecture is built on a streamlined and modern design, utilizing a deep neural network architecture based on the backbone of CSPDarknet53 or CSPDarknetLite (*Jocher, 0000*). It is trained end-to-end using a combination of the cross-entropy loss for class predictions and the generalized intersection over union (GioU) loss for bounding box predictions. The GIoU loss improves the bounding box localization accuracy and encourages better box shapes, leading to more accurate object detection results. Anchor boxes are boxes featuring a fixed aspect ratio in YOLO that are used to predict the class and positional offset of the bounding box. Anchor box coordinates consist of widths and heights that are frequently computed on the training dataset employing k-means clustering to accurately represent the dataset's most prevalent shapes. Choosing appropriate anchor boxes can significantly improve object detection speed and accuracy. The actual bounding box is calculated utilizing these predictions in the following manner:

**Center coordinates:**

$$b_x = \sigma(t_x) + c_x \text{ and } b_y = \sigma(t_y) + c_y.$$

Here, $b_x$ and $b_y$ are the absolute values that represent the centroid locations and $t_x$ and $t_y$ are the centroid location relative to the grid cell in x and y coordinates respectively. The $\sigma$ is the sigmoid function and $(c_x, c_y)$ is the top-left coordinates of the grid cell.

**Dimensions:**

$$b_w = p_w e^{t_w} \text{ and } b_h = p_h e^{t_h}.$$

Here, $b_w$ and $b_h$ represent the image's absolute width and height, respectively. Whereas, the $p_w$ and $p_h$ isthe anchor box obtained prior to clustering.

YOLOv5 yields three outputs: the classes of the detected objects, their bounding boxes, and objectness scores. The class loss and objectness loss are computed using binary cross entropy (BCE). Whereas, the complete intersection over union (CIoU) loss is used to determine the location loss. The final loss formula is represented by the following equation.

$$\text{Loss} = \lambda_1 L_{cls} + \lambda_2 L_{obj} + \lambda_3 L_{loc}.$$

## Hardware

The hardware component of our system, on the other hand, consists of a robotic arm for waste handling, a Raspberry Pi for loading the deep learning model (*Behera et al., 2020*), and an Arduino for managing the robotic arm's movements. The suggested approach makes use of the advantages of both software and hardware by utilizing robotic arm manipulation and AI-based image recognition. The primary processing unit used for image recognition tasks is an energy-efficient and portable device such as an Nvidia Jetson Nano or Raspberry

Pi. They make use of a computing architecture that is (GPU) accelerated, enabling quick and concurrent computations for real-time object detection. The CUDA GPU present in the Jetson Nano makes it perfect for deploying sophisticated machine-learning models in environments with limited resources. RPi is also a good option for integrating the system's hardware and software due to its small form factor, low power usage, and General Purpose Input/Output (GPIO) pins. Thus, the system was initially developed using the Pi3 and later deployed on the Jetson Nano because of its higher computational performance. Finally, instructions are communicated to the Arduino, which is apt for controlling low-level robotic actions due to its adaptability and simplicity.

The structural components of the robotic arm were meticulously designed and then produced using 3D printing technology. This method allowed for the creation of lightweight yet robust parts that could be easily customized to fit the specific requirements of the arm's design, such as mount points for the motors and channels for wiring. After the 3D printing process, the assembly of the robotic arm involved careful integration of the motors with the printed parts. The MG996R and SG90 motors were affixed at strategic points using a combination of screws and brackets, ensuring a secure and functional fit. The assembly process also included the installation of necessary wiring and control systems, which were routed through the arm to connect each motor to the central control unit. This setup allowed for coordinated movement of the arm's different segments, controlled either by a computer system or a manual interface as shown in Fig. 1.

## Dataset

The primary dataset, 'Garbage Object-Detection' (*Material identification, 2022*) is extracted from Roboflow, which is a software platform that allows image preprocessing, annotation, and augmentation, especially for computer vision research. The dataset comprises a total of 10,464 images from six different classes, namely Biodegradable, Cardboard, Glass, Metal, Paper, and Plastic. The data is split in the ratio of 7:2:1 for training, validation, and testing. The total size of the dataset is approximately 250 MB. Each image has its own associated annotated label in a text file, which contains information about the class label and the four edge coordinates of the bounding box of the waste items present in the images. The data.yaml file contains the path to the training and validation images and labels. This file will also include the class names from the dataset. To enhance the balance of our dataset classes, we incorporated an additional dataset, Alphatrash (*Tiyajamorn et al., 2019*), featuring 100 selected metal and plastic images each into our metal and plastic class categories, respectively. To train this dataset on object detection models, bounding box annotations were created around each metal and plastic object in all of the images. The dataset was uploaded to the Labelbox web app, which allowed for the generation of custom object detection datasets to generate bounding box annotations.

Figure 2 shows the high-level flow of the proposed YOLO single-shot detector interfaced with the robotic arm system. The proposed system makes use of the PyTorch Deep Learning Framework to train the custom model. A wide range of tools and features are available in PyTorch for building and using neural networks. It is a well-liked option for machine learning research and development due to its user-friendly interface, thorough

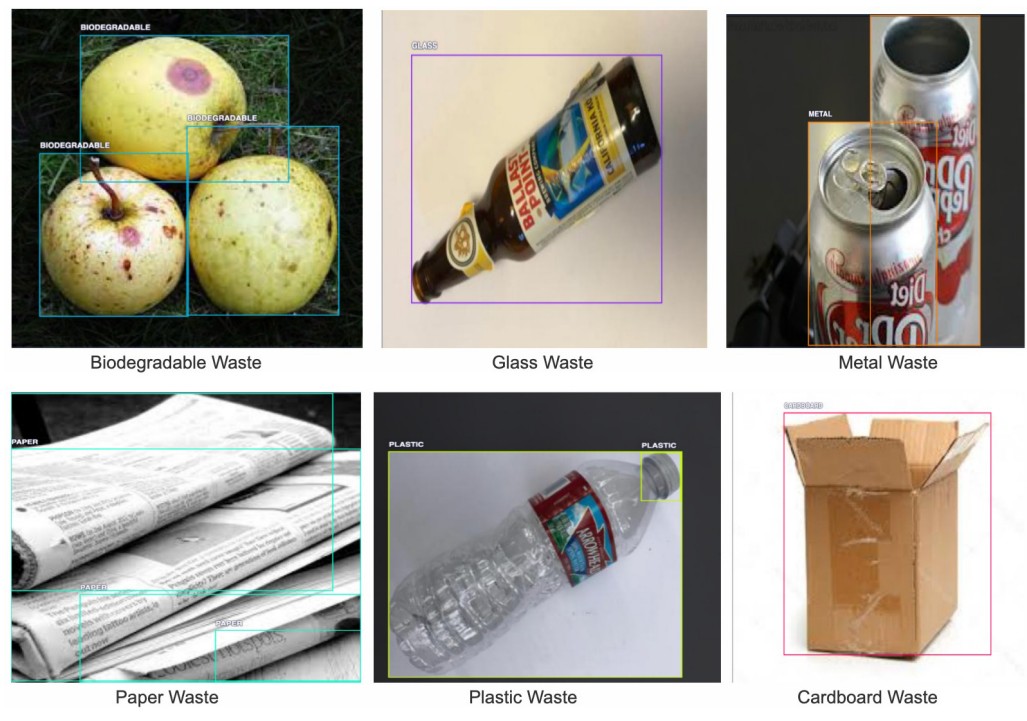

**Figure 1** **An example of each class present in the utilized dataset, namely biodegradable, glass, metal, paper, plastic, and cardboard waste types.**

documentation, and active community support. Also, the system can make use of cloud computing tools like Google Colab while it is training. This cloud-based approach promotes collaboration and scalability while doing away with the need for substantial local computing resources.

Overall, this suggested methodology provides a thorough and reliable solution for waste segregation, integrating cutting-edge technologies to produce the best outcomes. Modern tools like YOLOv5, Computer Vision, Nvidia Jetson Nano, Arduino, PyTorch, and Google Colab are utilized to achieve the desired result. This approach promises to significantly improve waste management systems by automating waste segregation procedures, which will increase sustainability and environmental conservation.

## System implementation

The system architecture of the proposed model is shown in Fig. 3. The system architecture mainly consists of four different modules based on their functionality:

 i. Video processing module
 ii. Waste localization and classification module
iii. Serial communication module
 iv. Robotic arm control

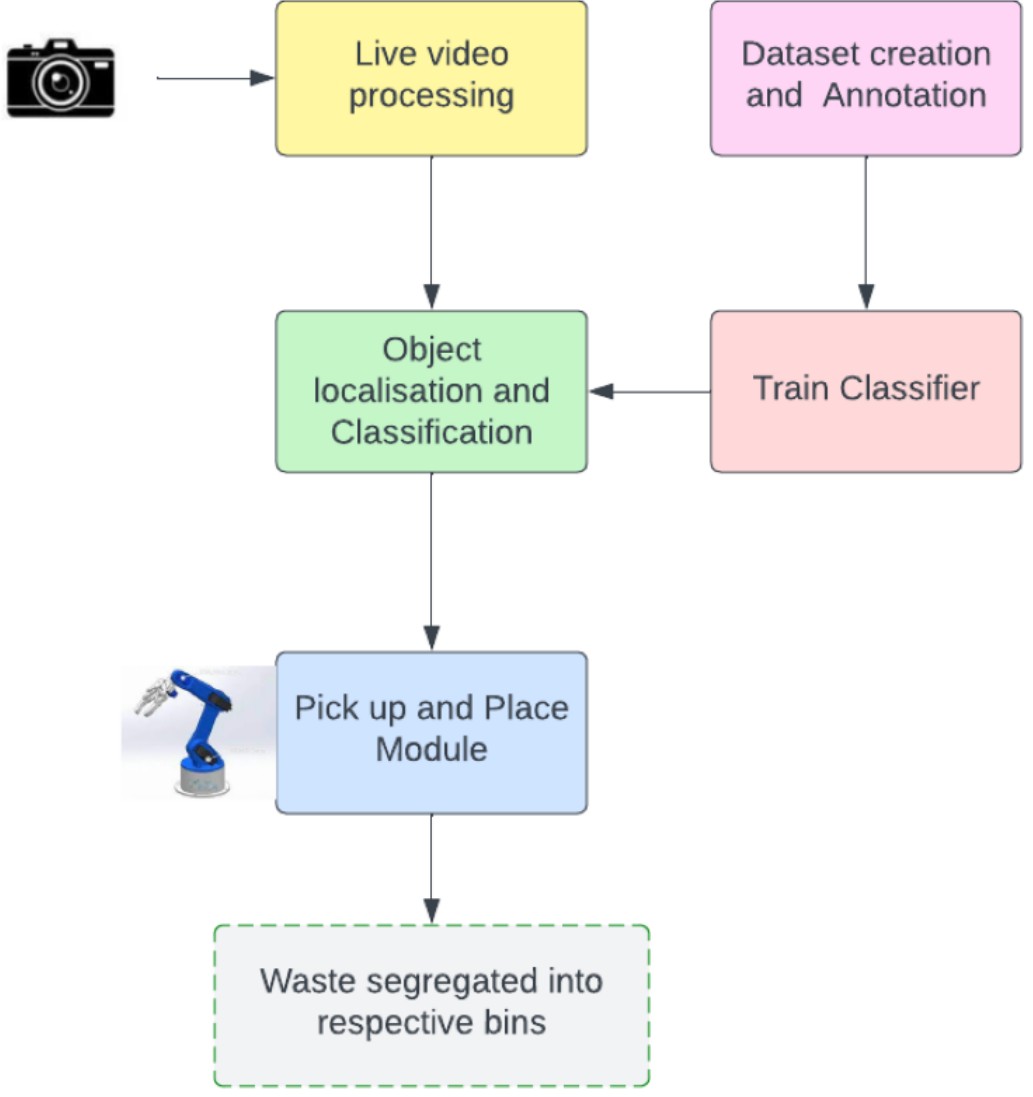

**Figure 2   High-level design of the proposed waste segregation prototype.** The design is broken down into modules in the flowchart, depending on the functionality.

### Video processing module

The video processing module facilitates obtaining the ground truth (GT) of the waste present in the experiment's ecosystem. An external camera sensor is used to stream live video of the intended waste items present in the ecosystem (*Zubair et al., 2022*). Following this, the video will need to be processed so that the deep learning model can carry out its detections as effectively as possible.

Image frames are extracted from the live video input of the external camera. The image frame is created as a snapshot of the video input every 5 s with the aid of the OpenCV library in Python; further, the image frames are resized to a width of 300 pixels. Each frame is analyzed, and the frame rate per second is monitored and recorded. The frame

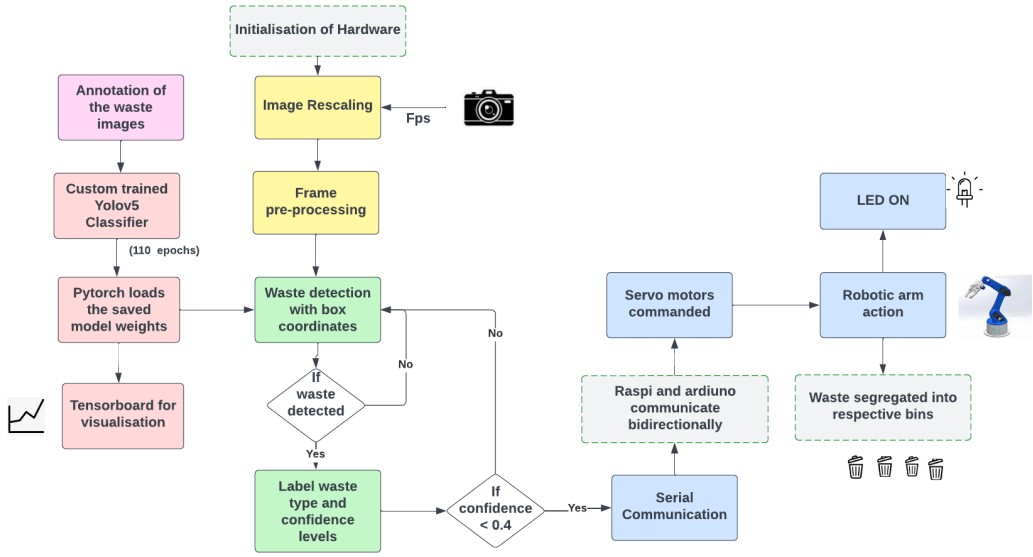

**Figure 3** **Comprehensive Architecture of the Proposed Robotic Arm System.** The detailed design and flow of our waste segregation prototype.

rate can be minimized depending on the CPU constraints, as it will allow the program to be more efficient and reduce the CPU load. A rescaling method is used on each frame, which treats each frame as a smaller image while processing for better efficiency. At this stage, the frame is encoded in BGR format, but for the convenience of waste classification, it will be transformed into RGB format and then PIL Image format using a numpy array. The PIL format is used because of its wide range of image modification functionality and metadata extraction from the image. The video that is collected from the camera sensor is continually analyzed on a frame-by-frame basis and then forwarded to the subsequent object localization and classification module.

## Waste localization and classification module

Object detection and classification are the primary responsibilities of the waste localization and classification module. This module uses the Yolov5 and OpenCV libraries. OpenCV provides a comprehensive set of image processing and computer vision capabilities, whereas YOLO focuses on the deep learning aspect of object detection. The Yolov5 model is trained using a publicly available data set containing six classes: biodegradable, cardboard, glass, metal, paper, and plastic. The images are annotated with bounding boxes that define the coordinates and dimensions of the visible objects. Each bounding box has a class label that specifies the category of the object. This dataset is processed, grouped in batches for faster training, and then trained over the Yolov5 architecture by introducing additional hidden neural layers to produce a customized SSD waste classification model. We train our model using three sets of data, *i.e.,* train, valid, and test. The open-source dataset is trained for around 120 epochs, which overall consumes approximately 15 compute hours. The trained model comprises approximately 113 neural layers and approximately 7.3 million parameters. To validate the robustness of the trained model, the 20% split from the (7:2:1 -

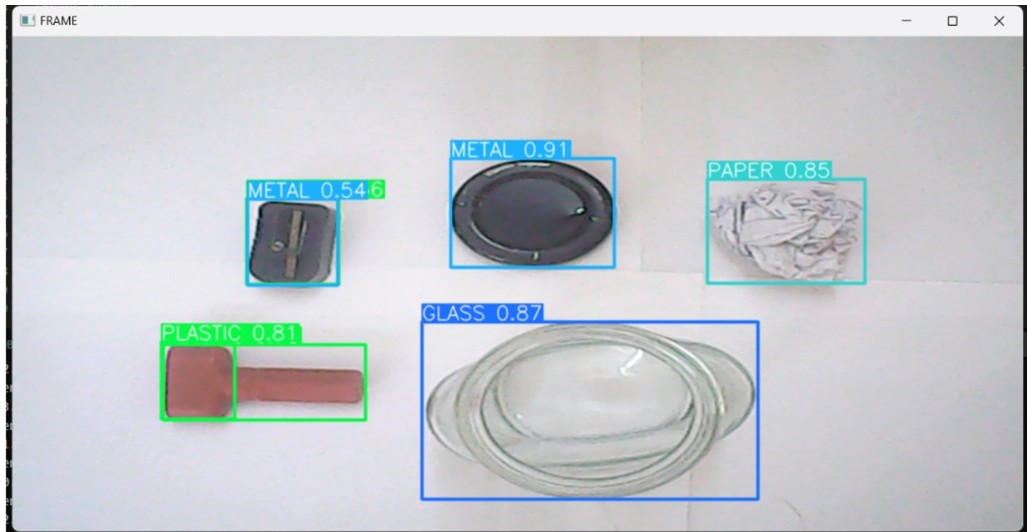

**Figure 4** **Object localization and classification of different waste classes captured in a real-time environment by the custom-trained YOLO model classifier.** Each detection is mapped to a boundary box, labeled, and displayed with its prediction confidence score (in %).

Train: valid: test) dataset is utilized. The validation is performed after each epoch, and the weights are optimized based on the performance of the model. This continuous evaluation of the model assists in better precision in fewer epochs. A constant frame size of 416x416 is used to train the model. The weights after each iteration of training are saved along with their parameters, and the best weights of the model are saved for further use in the serial communication module as shown in Fig. 4.

## Serial communication module

Serial communication is a widely employed method for device-to-device data exchange, and it is commonly used to transmit data between a Raspberry Pi and an Arduino. Utilizing the Universal Asynchronous Receiver-Transmitter (UART) protocol, the Raspberry Pi and Arduino can establish a serial connection.

The UART protocol enables serial interface data transmission and reception. It consists primarily of a transmitter (Tx) and a receiver (Rx). The transmitter converts parallel data to a serial bit stream, whereas the receiver converts serial data to a parallel format. To establish serial communication between a Raspberry Pi and an Arduino, the Tx and Rx pins of both devices must be connected. The data transmitted over the dedicated pins can also be transmitted *via* the USB's data pins. This simplifies the process of setting up the environment, reduces the risk of data loss, and enables the use of libraries for multiple programming languages. Python's serial library allows us to utilize this feature. Consequently, a USB A-to-USB B high-speed data transmission cable is used to connect the RPi and Arduino.

The Raspberry Pi's built-in serial interface, typically referred to as "/dev/ttyAMA0" or "/dev/serial0", is used for serial communication. Python (3.6+) is used to communicate

with the serial port on both the Raspberry Pi and Arduino *via* the aforementioned USB connection (Rx/Tx cables would perform similarly). Libraries such as pySerial and Serial are used on the Raspberry Pi to send and receive data over the serial connection. These libraries offer a straightforward and convenient method for establishing serial communication as well as functions for reading and writing data without a hitch.

On the Arduino side, the serial library included with the Arduino IDE can be utilized. Using familiar functions such as Serial.begin(), Serial.print(), and Serial.read(), this library enables sending and receiving data over the serial port. To exchange data between the Raspberry Pi and Arduino, a communication protocol can be defined that specifies the format and structure of the messages being sent. For example, it is possible to delimit each message with a start marker, data payload, and end marker. To ensure proper data transmission and interpretation, the Raspberry Pi and Arduino must both adhere to the specified protocol.

Here, once the GT input frame from the camera is processed by the YOLOv5 model to detect waste, the output from the model is used for communicating the directives to the robot. Based on the detected "class" or "id" by the model, the developed model instructs the Robotic arm to sort the waste must be sorted. The Python code that reads the previously mentioned class that the model identified contains a switch/if case. Based on the class it falls under, a command is sent to the Arduino to move the robot accordingly. On receiving the "start" signal for picking up the object from the serial communication, the robotic arm extends over its multi-axis arm and picks up the detected waste. The waste detected is assumed to be at the center of the screen, and based on the predefined mathematical calculations, the robot moves the arm to pick up the waste from the center of the screen. Once the waste is picked, based on the directives that were received from the waste classification model, it is dropped at its respective sorting zone. After dropping the waste in its designated zone, the completion flag is relayed back from the Arduino to the Raspberry Pi. This completes a circular sync of the program, and the iterations are recorded. The robotic arm returns to its original position to continue the same process until the program is interrupted.

## Robotic arm control

The robotic arm is a critical aspect of the waste segregation environment. Once the model receives information from the serial communication port on the Arduino module, the pre-written code in the microcontroller can control the robot on its five different axes. This controllable axis consists of multiple interconnected segments or links, allowing it to move in numerous directions. Each joint of the robotic arm is equipped with a servo motor. Servo motors are commonly used in robotic systems due to their precise control over position and angle. The servo motors can rotate the arm segments to different degrees, enabling the arm to reach various positions and orientations.

The robotic arm requires both data and power connections to operate effectively. The data connection, likely through cables or wireless communication, allows the arm to receive instructions and commands from a control system or a computer program. The power supply provides the necessary electrical power to drive the servo motors and other

components of the arm. In the previous segment, the article discussed how the Python code instructs the Arduino *via* serial communication to sort the waste based on detection. Once the Arduino receives which segregation "bin" the waste must be put into, each of the 5 motors within the robotic arm is moved sequentially to perform the task. This is accurately possible without the need for any external monitoring sensors for the robotic hand to know its position using servo motors. These servo motors are programmed to know their current degree, which lets us know exactly where the robot is and also enables the program to command the robot to go to a particular spot easily.

A one-axis gripper is equipped with the robot to pick up the waste. The Arduino is in charge of controlling this gripper to pick and place objects as needed. The gripper is padded with velcro or rubber to enhance its grip, ensure the secure handling of the waste items, and prevent dropping when slippery. The body or structure of the Robotic arm is created using 3D printing technology. 3D printing allows for the precise manufacturing of complex geometries and customized designs. The 3D-printed body provides the necessary strength, rigidity, and lightweight construction for the Robotic arm.

## Experimental setup

The experimental setup made to check the workings of the proposed multi-class waste segregation using computer vision and a robotic arm consists mainly of a robotic arm, a Burette stand, waste bins, an Arduino, a Raspberry Pi, and jumper wires. The Burette stand is used to place the camera in a still position and is capable of capturing a top view of the ecosystem. The servo motors present in the robotic arm are connected to Arduino pins and external 5V and ground. In this way, Arduino can command the individual servo motors to perform the required rotation needed for the overall correct movement of the robotic arm (*Material identification, 2022*). The Raspberry Pi is connected to the Arduino through USB for serial communication between the devices.

An open-source dataset containing images of different waste classes is imported and processed for inference and training. This collected dataset consists of six different waste classes: biodegradable, cardboard, glass, metal, paper, and plastic. The dataset was annotated using online tools, and each image has a text file associated with it that includes the label and box coordinates of the waste item present in the image. This is then worked upon to create a machine-learning model capable of distinguishing and locating any waste classes in the input image frame received from the still camera. By extensively training the YOLOv5 model using the training dataset, model weights are precisely fine-tuned to be capable of accurate waste classification for real-world waste images. The result from the model is then used to instruct the hardware segment of the proposed system to perform actions based on the class of waste that is present. The required parts for the robotic arm are 3D printed and assembled with the required motors. This movable arm with a fixed base will be controlled using Arduino to grab, hold, move, and drop the waste item.

In the experiment setup shown in Fig. 5, we conduct a randomized real-life test case scenario to assess the precision of the model and the overall system in relation to the environment and the objects. In our robotic arm system, we concentrate on the physical separation of four primary categories of dry waste: paper, glass, metal, and plastic, even

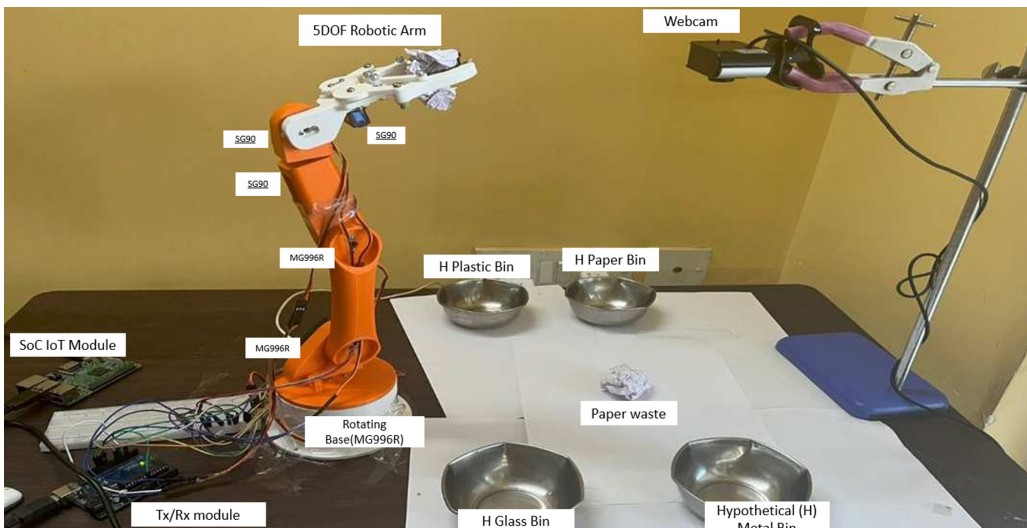

**Figure 5** **Experimental setup –frames from the webcam (right) are processed by the trained model in the Raspberry Pi, and the waste type is classified.** The data is passed onto the robotic arm (left) and segregated into the correct bin.

though the model is trained on six different classes In order to conduct an experiment, we obtained various categories of waste commonly found in the garbage of an average household on a regular basis. This encompasses a variety of colors, and forms and also contains transparent, translucent plastic and glass. The various specimens of waste are placed together in a single container and mixed randomly without being seen. Input waste material was placed in the test environment and was chosen by picking a random waste material from this collection box for many iterations. We were able to achieve an average success rate of eight out of 10 classifications. The primary instances in which the classifications predominantly faltered occurred when the model experienced difficulty distinguishing between plastic and glass.

## RESULTS AND DISCUSSION

The YOLOv5 model is evaluated primarily on the following evaluation metrics: precision, recall, and the three losses: box loss, classification loss, and objectness loss. The trained YOLOv5 model has been validated using measures such as precision and recall to access the model's proportion of true positives and actual positives that are correctly identified. The three losses, on the other hand, aid in determining the YOLOv5 model's accuracy of the location and classification of the object.

The system's speed and compactness are unique aspects of our system. The fundamental component of the system is a bespoke YOLOv5 model, renowned for its rapidity and effectiveness in identifying objects. This model is designed specifically for precise garbage sorting and is connected to an adaptable 5DOF (Degrees of Freedom) robotic arm. The 3D-printed robotic arm with precise servo motors serves as an example of the article's hardware and software incorporation. The notable aspect of this study is its pragmatic

implementation in a real-life setting, showcasing an average efficacy of the overall system of 80% for garbage categorization. Previous studies in this domain explore mainly earlier versions of YOLO, and this study of automated waste management utilizes the advanced YOLOv5 model, which can operate on SoC IoT devices like Raspberry Pi, in conjunction with a well-crafted robotic arm for waste segregation. In addition, we not only used a single system-on-chip (SoC) device for image classification, but we also separated the major functionalities by employing serial communication. This approach delegates the task of handling robotic arm movements to an Arduino. This division is more effective and simpler because the Arduino is well-suited for controlling hardware movements, while the SoC can focus on computationally intensive image classification. This system not only represents environmental sustainability but also showcases the capacity of AI and robotics to address practical, real-world problems.

## PRECISION AND RECALL

Both the accuracy terminologies, precision, and recall can be seen consistently improving over the epochs for the training and validation datasets. The model was able to achieve a maximum of 0.65 precision and 0.5 recall, and this can be further improved if trained for more epochs in a better GPU environment with limited constraints as shown in Fig. 6.

### Box, classification, and objectness losses

Three types of losses have been measured while training the model, as represented in Fig. 7. The box loss represents how well a machine learning technique can locate the centre of an object, in addition to how effectively the predicted bounding box covers a given object. Classification loss indicates how accurately the YOLOv5 model can predict the proper class of a given object. Objectness loss resulting from an incorrect box-object intersection over union (IoU) prediction instructs the network to accurately predict the IoU. The graph of all three losses is consistently decreasing for both training and validation data, indicating a decent AI model with good learning and results. Overall, a waste segregation system that combined both hardware and software technologies was successfully built and tested.

### Confusion matrix

The confusion matrix is an effective metric in the context of object identification since it gives a thorough and complete evaluation of an object detection model's performance. It is very useful for testing a model's predictions on a dataset with the objective of identifying and locating objects within images. The confusion matrix is a table that summarises the success or failure of a classification system by categorizing and organizing its predictions.

The confusion matrix for object detection is more nuanced than for image classification. The overlapping region of each forecast, as well as the confidence score, are taken into consideration. Figure 8 displays the confusion matrix of our trained model. The results of the true positives and true negatives are promising for the five classes: biodegradable, glass, metal, plastic, and cardboard which is evident in the confusion matrix. However, the results for the paper class are significantly lower since the model misclassifies the paper as background (no object detected in the image). This is mainly due to the paper's thin and

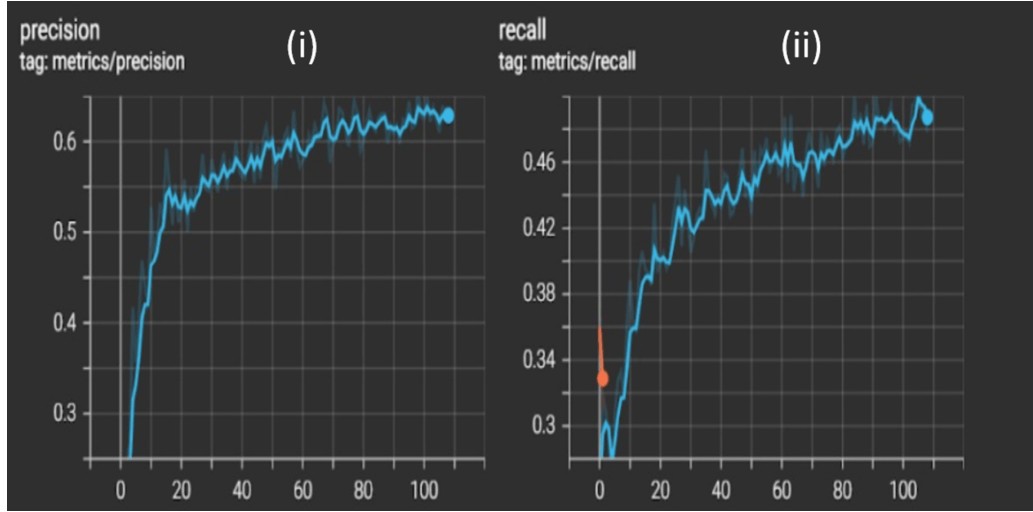

**Figure 6** **Graphs of trained YOLO classifier metrics (i) precision (ii) recall where the $x$-axis is epoch number and the $y$-axis is the metric value.** An increasing graph depicts that the model is learning the features successfully and is improving its prediction capabilities.

flat nature, which makes it harder for the model to differentiate it from the background. The limited range in the training dataset, such as the lack of variations in color, texture, and background, also contributes to the paper's frequent incorrect classification. Despite the model's lower accuracy in identifying paper in validation and test sets, this did not have any significant impact on our overall robotic arm segregation system's effectiveness in real-time experiments. Our prototype in the test environment used a dark contrast background for the placement of waste items. In consideration of the arm's gripper and grasping capability, crumpled papers were used as the test subjects rather than flat papers. This setup allowed for the accurate classification of papers, overcoming the challenges observed in the model. To enhance paper detection accuracy, this study suggests expanding the training dataset with diverse paper forms, lighting, and backgrounds and implementing data augmentation for robustness against variable conditions.

## CONCLUSIONS

The findings of this study highlight the potential advantages of implementing waste management systems driven by artificial intelligence. The final prototype of the robotic arm-based waste segregation system could classify and drop test waste items 80% successfully into their respective containers. Using a custom fine-tuned YOLOv5, a robust object detection framework, in conjunction with hardware components such as the Arduino and Raspberry Pi, enabled accurate waste classification, precise robotic arm control, and seamless data communication. The three losses calculated for the trained model on the garbage dataset were box loss, class loss, and object loss at 0.047, 0.078, and 0.0392 respectively, at around 100 epochs. The significant misclassification of paper as background resulted in the overall precision of the trained model being 0.65. While the paper class caused issues, the other

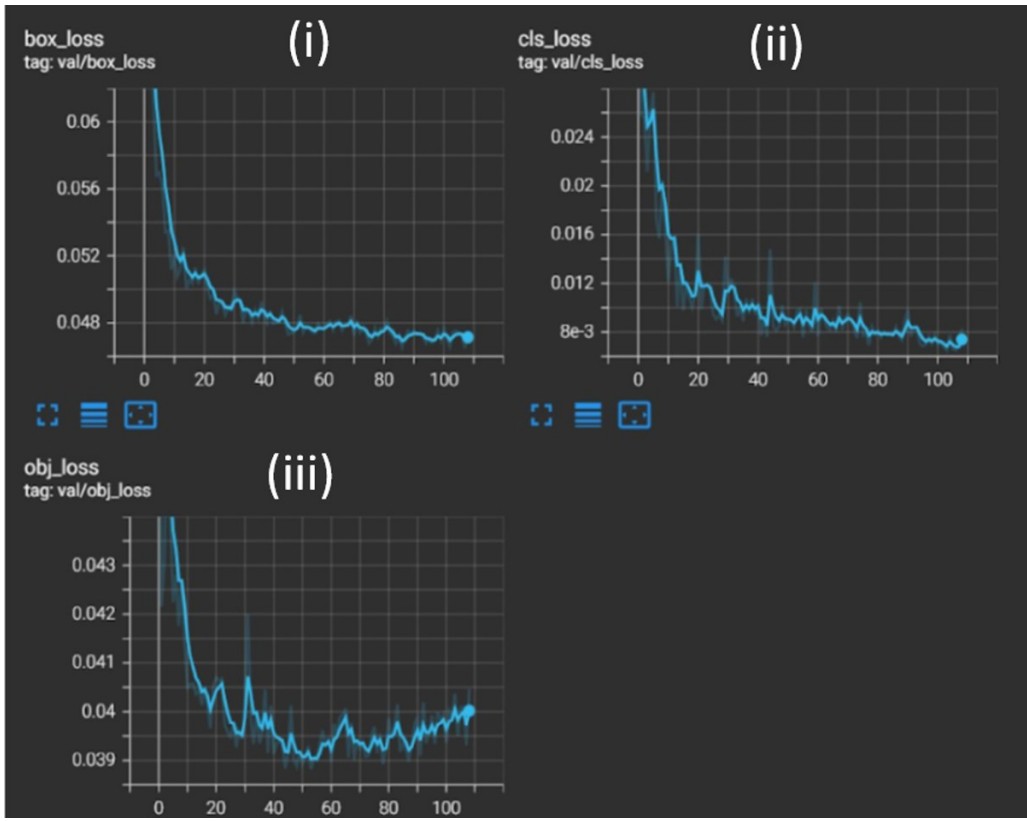

**Figure 7 Three box losses of the custom-trained YOLO model, namely (i) box loss, (ii) class loss, and (iii) object loss.** They are analyzed to understand the training progress of the YOLO model. The loss graphs of the validation set are represented, where the $x$-axis is the epoch number and the $y$-axis is the respective loss value. All three losses have a decreasing curve, signifying consistently improved learning of the model. (i) Box loss $-0.047$ at $\sim$100 epochs. (ii) Class loss $-0.0078$ at $\sim$100 epochs. (iii) Object loss $-0.0392$ at $\sim$90 epochs.

categories performed exceptionally, with the glass class showcasing the best performance. It achieved a precision rate of 0.92, indicating high accuracy in correctly identifying glass items. Additionally, it exhibited a recall of 0.72, reflecting a substantial proportion of actual glass instances being correctly detected by the model. In future work, the focus will be on enhancing paper classification accuracy by training the model on a much bigger diverse dataset. Additionally, the model's training was constrained by GPU limitations, resulting in fewer epochs. Increasing the number of training epochs to optimal value is anticipated to improve the model's overall confidence and accuracy, particularly in classifying complex and mixed waste items. This extended training period is expected to refine the model's learning process, leading to enhanced performance in waste segregation tasks.

This study has examined the importance of waste management and the significance of waste segregation in achieving sustainable and effective waste disposal practices. By leveraging technologies such as computer vision, machine learning, and hardware-based robotic arms, waste segregation processes can be automated, resulting in enhanced

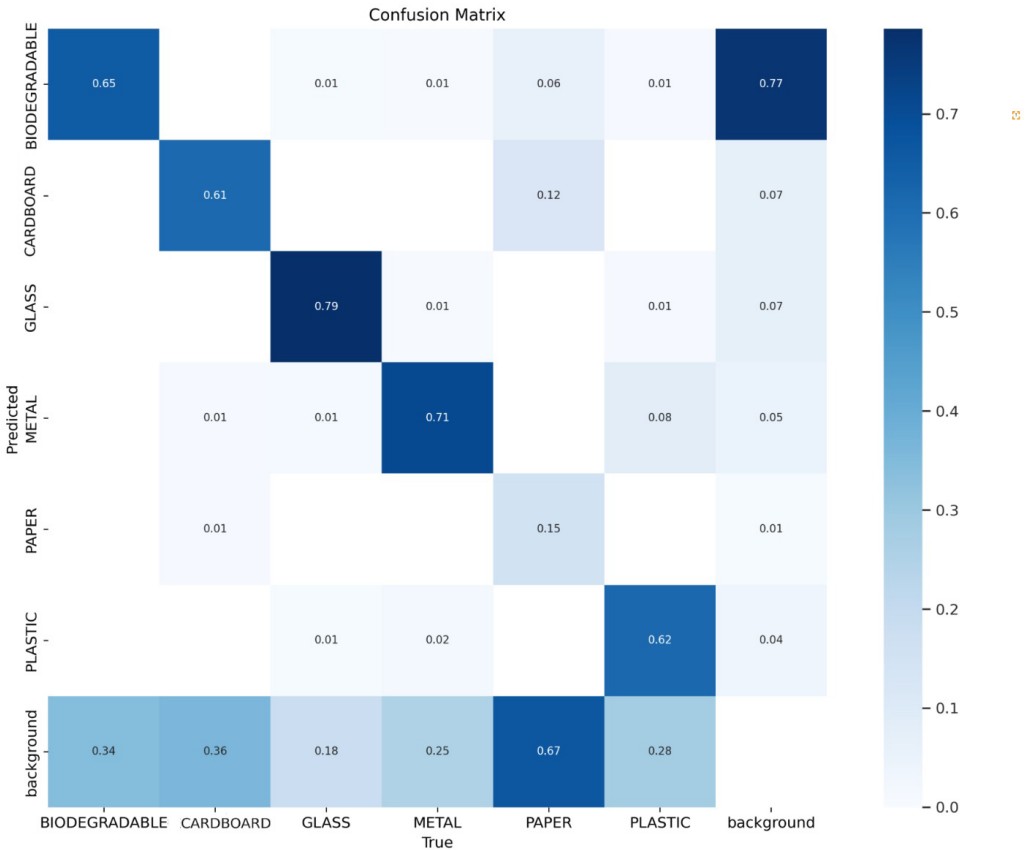

**Figure 8** **Confusion matrix showing the actual prediction on the _X_-axis and the model's prediction on the Y-axis.** The matrix has seven rows and columns each, which includes the biodegradable, glass, metal, paper, plastic, and cardboard waste types and the background.

precision, efficiency, and overall efficacy. Education and community involvement are essential for fostering a culture of waste segregation and recycling, as they encourage individuals, households, businesses, and governments to participate actively. This article concludes by emphasizing the critical need for continued research, collaborative efforts, and practical implementations of AI-driven waste management systems for a better future for the environment.

### Funding
The authors received no funding for this work.

### Competing Interests
The authors declare there are no competing interests.

## Author Contributions

- Jayanti Lahoti conceived and designed the experiments, performed the experiments, analyzed the data, performed the computation work, prepared figures and/or tables, authored or reviewed drafts of the article, and approved the final draft.
- Jathin Sn conceived and designed the experiments, performed the experiments, analyzed the data, performed the computation work, prepared figures and/or tables, authored or reviewed drafts of the article, and approved the final draft.
- M. Vamshi Krishna conceived and designed the experiments, analyzed the data, prepared figures and/or tables, authored or reviewed drafts of the article, and approved the final draft.
- Mallika Prasad conceived and designed the experiments, analyzed the data, prepared figures and/or tables, authored or reviewed drafts of the article, and approved the final draft.
- Rajeshwari BS conceived and designed the experiments, analyzed the data, prepared figures and/or tables, authored or reviewed drafts of the article, and approved the final draft.
- Namratha Mysore conceived and designed the experiments, analyzed the data, prepared figures and/or tables, authored or reviewed drafts of the article, and approved the final draft.
- Jyothi S. Nayak conceived and designed the experiments, analyzed the data, prepared figures and/or tables, authored or reviewed drafts of the article, and approved the final draft.

## Data Availability

The original dataset is available at Roboflow:

https://universe.roboflow.com/material-identification/garbage-classification-3/dataset/2.

The waste data is available at Zenodo: Jayanti Lahoti. (2024). jayantiii/WasteDetection: Multi-Class Waste segregation using AI ad robotic arm (latest). Zenodo. https://doi.org/10.5281/zenodo.10604554.

The code is available at Zenodo: Lahoti, J. (2024). Waste Detection AI model file. Zenodo. https://doi.org/10.5281/zenodo.10876217.

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

.