# Peer review of "Multi-class waste segregation using computer vision and robotic arm"

_PeerJ Computer Science, doi:10.7717/peerj-cs.1957_

## Round 0.1 · original submission · Major Revisions

Please revise the paper according to the reviewer's comments.

**Language Note:** The review process has identified that the English language must be improved. PeerJ can provide language editing services - please contact us at [email protected] for pricing (be sure to provide your manuscript number and title). Alternatively, you should make your own arrangements to improve the language quality and provide details in your response letter. – PeerJ Staff

·

Basic reporting

Dataset details need to be clearly mentioned or more details required about dataset.

Experimental design

Research topic is good and has lot of applications.

Validity of the findings

Results are fine, however the evaluation metrics considered need to be specified.

Additional comments

1. This paper discuss about Multi-class waste segregation using YOLO single shot detector and using robotic arm sysyem
2. List of keywords missing
3. In abstract, add proposed model name
4. What was the training-testing ratio of classifier
5. How the model evlauated? What were the evaluation metrics for the proposed model?
6. Why PIL format?
7. Results section, yolo model performance metrics were missing
8. Dataset section can be added, specifying the number of images, image size etc can be specified

Cite this review as

Reviewer 2 ·

Basic reporting

It seems like there are several key issues with the study that need to be addressed to improve its quality and increase the likelihood of publication. Let's elaborate on each concern:
Abstract lacks exact results:
The abstract should provide a concise summary of the study's objectives, methods, results, and conclusions.
Include specific numerical findings or key outcomes to give readers a clear understanding of the study's significance.
Missing dataset description in the Methods section:
Clearly outline the characteristics of the dataset used in the study, including its source, size, variables, and any preprocessing steps applied.
A comprehensive dataset description is crucial for transparency and reproducibility.

Experimental design

Lack of data on proposed model validation:
Present details on the validation process for the proposed models, including the metrics used and the criteria for model selection.
Validation results contribute to the credibility of the study and help readers assess the reliability of the proposed models.
Limited discussion and lack of comparison to existing studies:
Expand the discussion section to delve into the implications of the results and their significance in the broader context of existing literature.
Compare the findings to relevant studies in the field to highlight the study's contributions and limitations.

Validity of the findings

Conclusion missing exact results:
The conclusion should reiterate the key findings and their implications.
Include specific numerical results to reinforce the study's main contributions and provide a clear takeaway for readers.
Limited explanation of utilized algorithms:
Expand the section on the algorithms used in the study, providing a detailed explanation of their underlying principles.
Include references to relevant literature or documentation for readers seeking a deeper understanding of the employed algorithms.
Figures captions lack exact details:
Ensure that figure captions provide sufficient information for readers to interpret the visual elements.
Include details such as axis labels, units, and any pertinent information needed to understand the figures without referring to the main text.

Additional comments

Addressing these concerns will not only enhance the clarity and completeness of the study but also increase its chances of being accepted for publication in its present form. It's essential to prioritize transparency, thoroughness, and engagement with existing research in the field.

Cite this review as

Reviewer 3 ·

Basic reporting

The basic informations are given very clearly.

Experimental design

1. In Figure1, how the dataset creation and annotation are carried out?
2. Input from camera is live video processing. how the video is convered into images and processed further.
3. What are the different types of frames are taken for pre-processing of frames? Kindly brief it.

Validity of the findings

The model can be compared with existing model and then validate the proposed result is good.

Additional comments

NIL

Cite this review as

---

## Round 0.2 · accepted · Accept

According to the comments of reviewers, and after comprehensive consideration, it is decided to accept it.

·

Basic reporting

author made changes based on review comments

Experimental design

experiment is good.

Validity of the findings

valid results

Additional comments

author made changes based on review comments

Cite this review as

Reviewer 3 ·

Basic reporting

The review comments are incorporated in the revised manuscript

Experimental design

NIL

Validity of the findings

NIL

Additional comments

NIL

Cite this review as